# Airborne influenza virus shedding by patients in health care units: Removal mechanisms affecting virus transmission

Francis Hanna[1,2☯], Ibrahim Alameddine[2☯], Hassan Zaraket[3‡], Habib Alkalamouni[3‡], Mutasem El-Fadel[1,2☯]*

1 Department of Civil Infrastructure & Environmental Engineering, College of Engineering, Khalifa University, United Arab Emirates, 2 Department of Civil & Environmental Engineering, Faculty of Engineering & Architecture, American University of Beirut, Lebanon, 3 Department of Experimental Pathology, Immunology & Microbiology, Faculty of Medicine, American University of Beirut, Lebanon

☯ These authors contributed equally to this work.
‡ HZ and HA also contributed equally to this work.
* mutasem.elfadel@ku.ac.ae, mfadel@aub.edu.lb

**Data Availability Statement:** All relevant data are within the manuscript and its Supporting Information files.

## Abstract

In this study, we characterize the distribution of airborne viruses (influenza A/B) in hospital rooms of patients with confirmed infections. Concurrently, we monitored fine particulate matter ($PM_{2.5}$ & $PM_{10}$) and several physical parameters including the room air exchange rate, temperature, and relative humidity to identify corresponding correlations with virus transport and removal determinants. The results continue to raise concerns about indoor air quality (IAQ) in healthcare facilities and the potential exposure of patients, staff and visitors to aerosolized viruses as well as elevated indoor PM levels caused by outdoor sources and/or re-suspension of settled particles by indoor activities. The influenza A virus was detected in 42% of 33 monitored rooms, with viruses detectible up to 1.5 m away from the infected patient. Active coughing was a statistically significant variable that contributed to a higher positive rate of virus detection in the collected air samples. Viral load across patient rooms ranged between 222 and 5,760 copies/$m^3$, with a mean of 820 copies/$m^3$. Measured $PM_{2.5}$ and $PM_{10}$ levels exceeded IAQ daily exposure guidelines in most monitored rooms. Statistical and numerical analyses showed that dispersion was the dominant viral removal pathway followed by settling. Changes in the relative humidity and the room's temperature were had a significant impact on the viral load removal. In closure, we highlight the need for an integrated approach to control determinants of IAQ in patients' rooms.

## Introduction

Indoor air quality (IAQ) is associated with serious health implications, with some facilities, such as hospitals, being more critical than others particularly in the presence of vulnerable patients. Physical and chemical characterization of IAQ in hospitals have been widely reported [1–8] with recent efforts targeting bioaerosols, especially viruses [9–16] such as the respiratory

**Funding:** The author(s) received no specific funding for this work

**Competing interests:** The authors have declared that no competing interests exist

syncytial virus (RSV) and influenza A/B that are the main causes of respiratory infections requiring hospitalization [17, 18]. Influenza virus can spread indirectly through contact with contaminated surfaces or via respiratory droplets and aerosols, particularly in crowded areas [17, 19–24]. The debate regarding the airborne transmission of respiratory viruses have intensified with the emergence of the avian influenza viruses and MERS-CoV [25], with concerns over airborne infections resurfacing with the COVID-19 pandemic [26–29] caused by the severe acute respiratory syndrome coronavirus (SARS-CoV-2).

Bioaerosols carrying viruses can reportedly remain suspended (<5μm) in the air for a few hours [30, 31], with two basic transmission routes for airborne viruses including droplets that are expelled through coughing and sneezing or through suspended aerosols (<5μm). Transmission by droplets is highly effective over short distances (<1m), while dispersal by aerosols can cover longer distances (>1m) [32]. Noteworthy, the "5μm" threshold between droplets and aerosols is not definitive and can be misleading [33, 34]. Aerosol particles ranging between 5 and 100 μm can reportedly also remain suspended up to 5.4 minutes before depositing within a few meters from the shedding source [34–38]. Thus, suspended aerosol transmission can also result from bioaerosols with a diameter less than 100 μm rather than 5 μm [34, 39]. Variations in temperature (T) and relative humidity (RH) play an important role in the deactivation of viruses [40–42]. Similarly, changes in the air exchange rate (AER) affect the transport and exposure to air contaminants [43]. Albeit the existence of design criteria-specifications for various physical parameters [44], aerosolization, transmission and survival of emitted viruses remain a concern at hospitals [45–47].

In this context, virus-related IAQ studies have relied on statistical and probabilistic analyses to assess the variability observed in the distribution of aerosols and the associated exposure risks within the hospital environment [10, 11, 13, 14, 48–52]. Virus transmission is reportedly affected within certain ranges of relative humidity (RH) and temperature (T) [53–56]. Despite earlier efforts and until recently, many of these studies fell short of defining clear statistical relationships between changes to the concentration of the influenza virus (or other microorganisms) in patient rooms as a function of physical determinants (such as T, RH, AER) and the location of the patient in the room [52]. With regards to the transport of the influenza viruses (or other microorganisms) from a patient, most of the research has been theoretical and consideration for the understanding of the effects of T or RH on the transmission routes remains limited [8, 39, 57]. Recent concerns over the spread of the COVID-19 pandemic have pushed the experimental and modeling envelope further, with a plethora of new modeling applications (both statistical and numerical) [39, 58–65]. While these new applications are of great benefit in visualizing the flow patterns within a confined space, they invariably face similar calibration challenges when using field measurements of the simulated COVID-19 virus.

In this study, we monitor the viral load (influenza A/B), particulate matter ($PM_{2.5}$ & $PM_{10}$,), and physical parameters (AER, T, RH) within patient rooms for the purpose of assessing airborne virus shedding by patients in health care units using an integrated approach of coupling field measurements with statistical and numerical analyses. The study contributes towards filling a gap in the reported literature on influenza virus characterization in hospitals and identifying determinants of airborne viral transmission and removal mechanisms.

## Materials and methods

### Study design

The study was approved by the Institutional Review Board (IRB) at the American University of Beirut Medical Center. All patients provided written informed consents prior to indoor air sample collection. As stipulated in the IRB approval, the research team did not recruit

participants. The team was only informed of the date a patient was diagnosed with influenza A/B. Otherwise, the healthcare team kept the identity of patients confidential. Research activities were limited to in-room monitoring of target indicators using well-defined and declared equipment with no obstruction of routine medical care procedures and hospital protocols.

Air samples were collected from single patient rooms that were selected after receiving laboratory-confirmation that the patient was infected with influenza. The Coriolis μ Biological Air Sampler (Bertin Instruments, France) was used in the sampling procedure. The Coriolis μ uses cyclonic technology, coupled with high suction rate (up to 300 L/min), to collect and concentrate virus-laden aerosols in a 15 mL liquid sample. Unlike other biosamplers, the Coriolis μ does not require a pump and operates at a much higher flow rate. Air samples were collected at two locations within each room: one sample was at 0.5 m away from the patient head and the other at 1 m away (Fig 1). The choice of the two distances was based on the reported critical droplet transmission distance [15, 66, 67]. In one room, samples were collected at 1 m and 1.5 m to check whether RNA copies can be detected at greater than one meter. During the sampling process, the patients were instructed to look in the direction of the air sampler to better capture expelled viruses. The Coriolis instrument has a high suction capacity and thus can have a potential overlap in the sampled volume if used concomitantly with another sampler in small patient rooms. As such, samples at 0.5 m and 1 m were collected separately to ensure accurate and unbiased sampling. The same sampling order was adopted for all patient rooms, starting with the nearest distance first at 0.5 m and then moving away to a distance of 1 meter once the sampling at 0.5 m is completed. All samples were collected at the breathing zone level of 1.5 meters and at least 1 meter away from the room walls. The sampling equipment was placed away from the ventilation system inlets and outlets, to avoid sample loss or cross contamination. The sampling process started shortly prior to requesting from a patient to cough

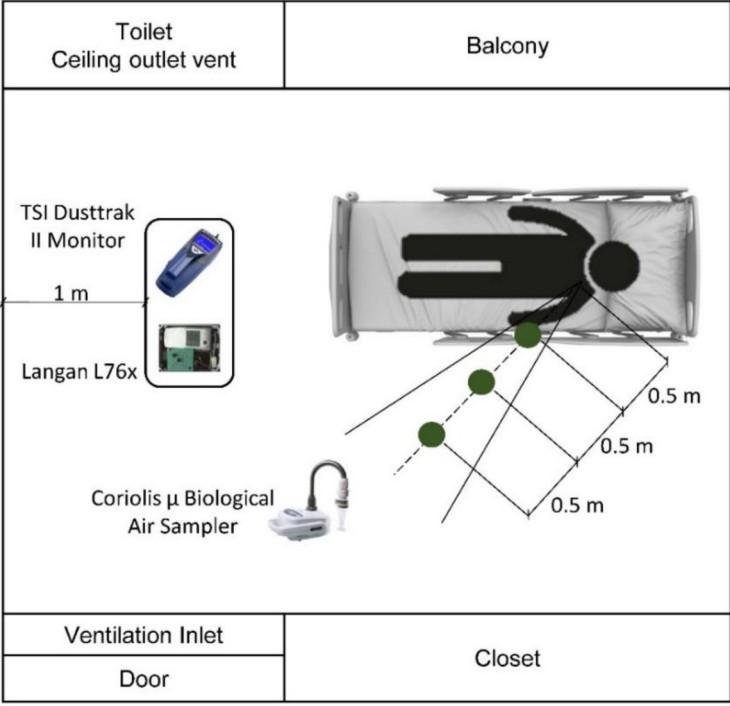

**Fig 1. Patient room layout and positioning of sampling equipment.**

to establish the base condition that would allow the assessment of the cough contribution to the initial concentration. Windows and doors were kept closed during the sampling process.

The air collected from patient rooms was aspirated for 10 minutes at a flow rate of 300 L/min and drawn into a collection tube containing 15 ml of sterile viral transport media (VTM). Noteworthy, the air sampler collects only aerosol particles ranging between 0.5 and 20 μm. The air sampler was decontaminated and air dried after each sample run to prevent potential carry-over contamination. The temperature (T) and relative humidity (RH) inside patients' rooms were also monitored and recorded over a period of 20 minutes using a portable Langan analyzer (model L76n), with a log interval of 10 seconds. Similarly, $PM_{2.5}$ and $PM_{10}$ levels were monitored using a portable TSI DustTrak II aerosol monitor (model 8532) with a log interval of 1 minute. During the monitoring period, occupancy levels (other than the patient) and the number of coughs were recorded to account for their potential effects on virus shedding, $PM_{2.5}$ and $PM_{10}$ concentrations. Occupants in the patient rooms were mainly nurses and doctors in addition to the patient. All occupants were advised to follow precautionary measures such as wearing face masks, gloves, disposable gowns, and were asked to sanitize before and after entering the patient room. This greatly minimizes potential virus shedding from sources other than the patient in the sampling area. Note that all rooms were deep cleaned prior to admitting a new patient and they all had a similar layout and size. The AER was also fixed throughout the sampling period across all rooms.

## Laboratory analysis

RNA extraction was performed on 250 μl of the air sample using the Purelink viral RNA/DNA Mini Kit (ThermoFisher Scientific) and eluted in 40 μl nuclease free water. Following extraction, 2 μl of the RNA extract was screened for Influenza A Virus (IAV) by probe-based quantitative reverse-transcription polymerase chain reaction (rt-qPCR) targeting the matrix gene (M gene) [68]. Positive samples had their copy numbers of the M gene estimated from the cycle threshold obtained from the rt-qPCR run, according to a standard curve correlating the logarithmic dilution of purified genome of known copy numbers (Vircell) to its cycle threshold (Ct) value. The number of IAV M gene copies per $m^3$ air was calculated using Eq 1.

$$\text{M gene } \frac{\text{copies}}{m^3} = \frac{\text{M gene copies } /\mu l \times \left(\frac{V_e \times V_m}{V_r}\right)}{U \times t} \tag{1}$$

Where
$V_m$ = volume of total media left after collection in $m^3$
$V_r$ = volume of specimen used for extraction in $m^3$
$V_e$ = eluted volume from the extraction in $m^3$
$U$ = collection flow rate in $m^3$/min, and T is the collection time in min.

## Statistical and numerical analyses

Indoor $PM_{2.5}$ and $PM_{10}$ levels in each room were averaged over the sampling period and compared with relevant IAQ guidelines [69, 70]. Additionally, the correlation (Pearson's) between the measured response variables ($PM_{2.5}$, $PM_{10}$, and RNA copies) and several potential predictors, such as temperature, RH, occupancy rate, and number of coughs, were determined. A steady-state Gaussian puff model [71] was also adopted and calibrated to assess airborne virus levels as a function of distance, shedding rate and frequency, AER, RH and T (Eq 2). The model was used to simulate the spatial trajectory of particles that move by advection, dispersion and settling. The model also included a transformation mechanism at the beginning of

the trajectory, whereby the emitted particles after coughing shrink from their original diameter to an equilibrium diameter $D_{eq}$ [72]. This phenomenon is a function of the ambient T and RH. At low RH levels, droplets are likely to lose more water than at higher levels. In this context, [73] developed a relationship to estimate the $D_{eq}$ as a function of T and RH, as well as the physiochemical properties of the droplets and the Kelvin effect (Eq 3). In this study, the change in diameter occurs immediately after a cough and the physiochemical properties of the emitted respiratory fluid was considered to include 8.8 g/L NaCl and 76 g/L of total proteins (TP) [72, 74]. Note that the size of the droplet affects the settling rate and removal efficiency as shown in Eqs 4 and 5.

$$c(x, y, z) = \frac{M}{(2\pi)^{\frac{3}{2}}} \sum\nolimits_{k=1}^{N} \frac{1}{\sigma_{xk}\sigma_{yk}\sigma_{zk}} \exp\left(-\frac{(x_k - x)^2}{2\sigma_{xk}^2} - \frac{(y_k - y)^2}{2\sigma_{yk}^2} - \frac{(z_k - z)^2}{2\sigma_{zk}^2} - \lambda t\right) \quad (2)$$

$$RH_{eq} = \exp\left(\frac{4\sigma M_w}{\rho RTD_{eq}} - \frac{M_w}{\rho_w\left(\left(D_{eq} - D_{m,s}\right)^3 - 1\right)} \sum \frac{v_y \theta_y \rho_y x_{s,y}}{M_y}\right) \quad (3)$$

$$S_i = \frac{2\rho g D_{eq}^2}{9\eta} \quad (4)$$

$$\lambda_{Settling} = -\frac{S_i}{H} \quad (5)$$

$$AER = \frac{AI \times 60}{v} \quad (6)$$

Where

c(X, Y, Z) = virus concentration (RNA copies/m$^3$)

M = exhausted viral mass (number of RNA copies) relative to each particle diameter

N = number of coughs

x, y & z = coordinates of the particles (m)

$\sigma_{x,k}$, $\sigma_{y,k}$, & $\sigma_{z,k}$ = x-, y- and z-directional deviation of the Gaussian distribution inside the $k^{th}$ puff, (m) [71] (Lateral deviations of the Puff model vary with the velocity $u_x$, which varies from one patient to another. $\sigma_y$ and $\sigma_z$ were estimated for every patient within a stepwise Monte-Carlo analysis aiming to minimize the mean square error)

λ = decay term due to settling and air exchange rate (1/minute) [74]

$(x_k, y_k, z_k)$ = spatial position of the $k^{th}$ cough [73]

$RH_{eq}$ = equilibrium RH for a specific droplet diameter ($D_{eq}$)

σ = surface tension of the droplet that is approximated by that of water = 0.072 Nm$^{-1}$

$M_w$ = molar mass of water = 18 g mol$^{-1}$

$M_y$ = molar mass of component y, composed of an inorganic fraction represented by salt (NaCl) and an organic fraction represented by total proteins (TP) ($M_{NaCl}$ = 58.4 g mol$^{-1}$, $M_{TP}$ = 66.5 x 10$^3$ g mol$^{-1}$) [74]

ρ = density of the entire droplet

$\rho_y$ = density of the y component (NaCl = 2165 Kg.m$^{-3}$, TP = 1362Kg.m$^{-3}$) [74]

R = ideal gas constant (0.0821 atm.L.K$^{-1}$.mol$^{-1}$)

T = temperature (K) [75]

$\rho_W$ = density of water (997,000 g.m$^{-3}$)

$D_{m,s}$ = mass equivalent diameter of a particle consisting of dry solutes (μm)

$V_y$ = stoichiometric dissociation number of component Y ($v_{NaCl}$ = 2, $v_{TP}$ = 1) [73]

$X_{s,y}$ = mass fraction of component Y ($x_{NaCl}$ = 0.104, $x_{TP}$ = 0.896) [72, 74]

$\theta_y$ = practical osmotic coefficient for component Y (TP = 3.75; NaCl = 0.95) [72, 74]

$S_i$ = settling velocity (m/second);

H = settling height = equal to the patient breathing level (H = 1.5 m)

$\eta$ = viscosity of the particle (g.m$^{-1}$.s$^{-1}$)

AER = air exchange rate (in 1/hour)

AI = air intake in (m$^3$/min)

V = room volume (m$^3$)

In this study, we assumed that all patients had the same coughed droplets diameter distribution as that reported in [76]. The amount of viral RNA remaining airborne over distance was calculated for each droplet diameter size before the total virus concentration was estimated. The particles were assumed to move at a constant velocity equal to that of the cough. Coughs were considered to occur at equal time intervals Δt. Moreover, viruses were assumed to be equally distributed across cough particles. While virus transmission can occur through droplets expelled by coughing and sneezing or via aerosols, this model only accounts for the short-range transmission due to the lack of sufficient data on deposition, resuspension and fractionation of infected aerosols, which can introduce significant bias. Note also that due to lack of data on the viability of the collected viruses, the biological decay of the shed viruses was not considered in the model. These assumptions are reasonable given the short time frame of the modeling exercise and the fact that the viral samples are cumulative over the entire sampling period.

Several variables were predefined in the model. These included the number of coughs, T, RH and AER. T and RH were the averaged values collected over the monitoring period. The AERs in this study could not be measured in each room due to imposed constraints in the presence of patients. AERs were calculated based on the air intake and the volume of each room (Eq (6)). In earlier work, the AER was measured in several similar rooms within the same hospital using a TSI ACCUBALANCE Air Capture Hood Model 8380 [77]. The measurements were then verified with the hospital's physical plant engineers. While the measurements occurred in a different time period, they provided confidence that the estimated AERs (7.2 air changes per hour) were close to those measured. The number of coughs was based on the actual number of patient coughs reported in each room. The surface tension for coughed droplets was initially set at 0.072 mN/m, the same as pure water, to reduce the number of calibration parameters. To assess the sensitivity of the results to changes in the surface tension, we varied this value based on the recent work of Vadlamudi et al., (2022) [78] who reported a surface tension of 0.0659 mN/m for coughed droplets. The results showed that within this range, changes to the surface tension value exhibited a negligible effect on the equilibrium diameter, and thus, a negligible effect on the settling efficiency.

The remaining variables that included the shedding rate, air flow velocities, lateral air velocities, and cough velocity were only defined by assigning statistical distributions to each based on literature reported values. The uniform distribution was adopted for most parameters, as it represents vague prior information on the parameter of interest. It also provides an upper and lower bound for each parameter. Similarly, lateral air velocities ($u_y$ and $u_z$) were assigned a uniform distribution with a minimum of 0.125 and maximum of 0.25 m/s [77]. Note that the ranges defined for the lateral air velocities were based on CFD simulations [77] on the same patient rooms. All patient rooms were assumed to have similar volumes, geometry and ventilation. As for the cough velocity ($u_x$), it was assumed to follow a uniform distribution with a minimum of 2.2 and a maximum of 22 m/s [79–81]. Finally, the shedding rate also follows a normal distribution with a mean of 15.8 and a standard deviation of 29.3 copies/cough [15].

Since the cumulative viral samples measured at 0.5 m and 1 m were not collected concurrently, the impact of having different number of coughs between the two samples needed to be attenuated. As such, normalized viral concentrations were used to calibrate the puff model. Normalization was conducted by dividing the total concentration measures in the sample by the number of coughs recorded over the sampling period. The model was executed and calibrated for all sampled rooms that had normalized concentrations above zero both at 0.5 m and 1 m. Calibrating the model for each room allowed us to account for the unique shedding characteristics of each patient (shedding rate and coughing speed).

A Monte-Carlo (MC) simulation was then used to calibrate the model. One million randomly generated combinations of shedding rates, air flow velocities, lateral air velocities, and coughing speeds were sampled independently and used to estimate the virus concentration at 0.5 m and 1 m. The combination of parameters that resulted in the minimal mean square error between the predicted and measured virus concentrations at 0.5 and 1 m away from the patient bed were selected and used to predict the virus concentrations at source (x = 0.1 m). Finally, the calibrated model was used to assess how varying T, RH and AER would affect viral transmission.

## Results and discussion

### Patient characteristics and virus detection

All patients that were isolated with droplet precaution and tested positive for IAV, influenza B were approached to volunteer in the sampling program. A total of 33 adult patients were enrolled and 65 air samples were collected. Twenty-eight patients tested positive for IAV, and 5 patients tested positive for influenza B (Fig 2). None of the patients was elderly. All patients were on anti-viral medical treatment. No information was collected on any other types of medication being administered. Patients were defined as emitters, if they had at least one air sample collected in their room testing positive for one of the viruses. Fourteen out of the 33 patients were found to be emitters (Fig 2).

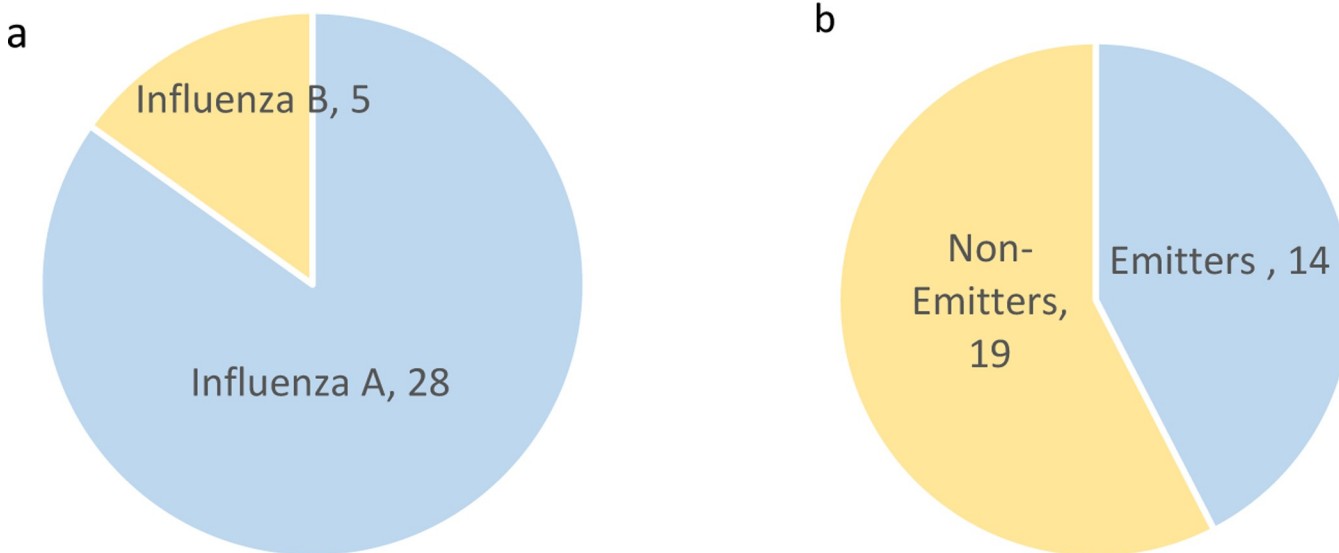

**Fig 2.** Characteristics of admitted patients (a) Distribution of patients by virus type; (b) Distribution of patients by virus shedding status. Emitters are patients with at least one air sample being positive for influenza virus.

Only air samples collected from IAV patient rooms tested positive for the virus (19 positive from the 55 IAV RNA samples collected). Meanwhile, none of the 10 influenza B RNA samples collected was positive. In total, viral RNA was detected in 19 of the 65 air samples collected. Six of the 19 positive samples (32%) were collected at a distance of 0.5 m away from the patient, 12 samples (63%) at a distance of 1 m, and 1 sample at a distance of 1.5 m (5%). Overall, the viral load across patient rooms ranged between 222 and 5,760 copies/m$^3$, with a mean of 820 copies/m$^3$ (Table 1). It should be noted that only adult patient rooms were sampled in the study. Interestingly, 68% of the positive samples were detected at a distance of 1 m or beyond from the patient suggesting that a significant fraction of the detected viral RNA is present on airborne particles and thus can travel away from the patient, posing a risk of airborne transmission. While the results are consistent with literature reported data indicating that influenza transmission may occur through large droplets traveling up to 1 m from the source [15, 67, 77, 82–84], they raise concerns that the current WHO and CDC safe distance recommendations (i.e. spacing of 1 m) may not be adequate to ensure the protection of visitors and healthcare practitioners during routine care operations in hospitals and similar facilities.

When the viral RNA concentrations were normalized by the number of coughs, we found that as the distance from the patient increased, the viral load exhibited a significant drop (Wilcoxon signed rank test, p = 0.03125). The normalized concentrations at 1 m were on average half of those measured at 0.5 m in the same room. In one room, where samples were collected at 1 and 1.5 meters, concentrations at 1.5 m were 8 times lower than those measured at 1 m.

The results revealed that active coughing was associated with a significantly higher positive rate of virus detection in the collected air samples (t-test; p-value = 0.049). Note that the discrepancy in the coughing rate explained largely the few instances when the concentrations measured at 1 m were higher than those measured at 0.5 m. Nevertheless, we still reported cases where we had two positive air samples even in the absence of any recorded coughs, highlighting the potential of the virus to remain suspended in the air for extended periods or to be shed by normal breathing and talking [82, 83, 85, 86]. IAV has been suspected to be transmitted by other mechanisms such as talking or simply breathing, which in turn expels fine infected aerosols [87]. For instance, and based on an observational study, Fabian et al. [67] reported that the virus might be contained in fine particles generated during tidal breathing. Similarly, Stelzer-Braid et al. [88] stated that the virus can be emitted during talking.

With regards to the association of the viral load with the physical environment in patient rooms, the normalized viral RNA load in the air samples were found to be positively, albeit not significantly, correlated with temperature (r = 0.43, p-value = 0.1). No significant correlations were found between the viral load and the measured PM$_{10}$ or PM$_{2.5}$ levels in the rooms. Our finding concurs with the conclusions made by Nor et al. [89]. Nevertheless, a previous study did report a positive correlation between PM and the influenza virus [90]. The link between PM and viral concentrations should be further explored. Also, there was no statistically significant correlation between the measured viral load and RH (r = -0.29, p-value = 0.2). Yet, the percent drop in viral load between 0.5 m and 1 m (ranged between 16.2% and 87.8%) was found to have a strong positive correlation with RH (r = 0.803; p-value = 0.036).

## PM and thermal comfort parameters

T and RH in patient rooms recorded an average of 23°C (20.6 to 25.4°C) and 48.4% (38.7 to 62.6%), respectively. These levels were largely within the American Society of Heating, Refrigerating and Air-Conditioning Engineers (ASHRAE) defined standard for hospital spaces [91]. Time-averaged PM$_{10}$ levels across patient rooms ranged between 9 and 89 °g/m$^3$, with a mean of 34 °g/m$^3$. As for PM$_{2.5}$, it ranged between 7 and 53 °g/m$^3$, with a mean of 24.5 °g/m$^3$.

**Table 1. Viral concentration measured in patient rooms with relative humidity, ambient temperature, and particulate matter.**

| Room ID Number | Distance (m) | Number of coughs | Viral Concentration (RNA copies/m3) | RH (%) | T (˚C) | PM10 (µg/m3) | PM2.5 (µg/m3) | Occupancy level |
|---|---|---|---|---|---|---|---|---|
| **617** | 1 | 18 | 1804.8 | 51.7 | 22.5 | 35 | 18 | 5 |
| | 0.5 | 0 | 339.4 | | | | | |
| **912** | 1 | 1 | 393.1 | 38.7 | 25.0 | 46 | 39 | 5 |
| | 0.5 | 0 | 0 | | | | | |
| **905** | 1 | 7 | 290.6 | 47.9 | 21.8 | 9 | 7 | 3 |
| | 0.5 | 4 | 0 | | | | | |
| **1041** | 1 | 8 | 0 | 42.8 | 22.6 | 21 | 13 | 3 |
| | 0.5 | 6 | 730.6 | | | | | |
| **527** | 1 | 5 | 0 | 40.8 | 24.0 | 19 | 15 | 3 |
| | 0.5 | 4 | 370 | | | | | |
| **526** | 1 | 12 | 298.8 | 44.0 | 25.0 | 26 | 17 | 3 |
| | 0.5 | 6 | 337.2 | | | | | |
| **915** | 1 | 1 | 221.9 | 45.5 | 22.5 | 19 | 9 | 3 |
| | 0.5 | 0 | 264.7 | | | | | |
| **1009** | 1 | 9 | 308.2 | 48.7 | 22.7 | 59 | 47 | 2 |
| | 0.5 | 8 | 0 | | | | | |
| **1012** | 1 | 30 | 224.8 | 45.0 | 21.1 | 59 | 45 | 2 |
| | 0.5 | 26 | 302.5 | | | | | |
| **1001** | 1 | 11 | 724.2 | 59.3 | 20.6 | 43 | 26 | 2 |
| | 0.5 | 4 | 0 | | | | | |
| **1007** | 1 | 7 | 290.5 | 51.0 | 23.4 | 20 | 14 | 2 |
| | 0.5 | 28 | 0 | | | | | |
| **1014** | 1 | 1 | 5760 | 48.7 | 25.4 | 12 | 10 | 3 |
| | 0.5 | 0 | 0 | | | | | |
| **924** | 1 | 3 | 575.4 | 42.7 | 21.7 | 78 | 42 | 2 |
| | 0.5 | 3 | 0 | | | | | |
| **1030** | 1 | 9 | 3225.6 | 62.6 | 23.5 | 89 | 36 | 3 |
| | 1.5 | 10 | 437.5 | | | | | |

RNA: Ribonucleic acid; RH: Relative humidity; T: Temperature; PM: Particulate matter.

Note that RH, T, PM, and occupancy level did not vary during the sampling period at both distances

Similar ranges were reported in the literature [92, 93]. Measured concentrations in several patient rooms exceeded international guidelines (Fig 3) which can be attributed to the resuspension of settled PM due to frequent visits or to the infiltration of outdoor $PM_{10}$ & $PM_{2.5}$ due to high indoor-outdoor PM correlation in the tested rooms [94]. Worth noting, no statistically significant correlation was detected between PM levels and occupancy rate.

## Numerical analysis

Several studies have attempted to simulate the shedding of the influenza virus, often focusing on the physical dispersion of the cough and neglecting the impact of T and RH on the virus dispersion [81]. More recently, much effort was dedicated to understand the effects of T, RH, and ventilation rates in the context of COVID-19 transmission in medical facilities [39, 58–60, 63, 95–97]. Fewer studies have highlighted the possible impact of humid and cold spaces on influenza virus survival or other symptoms [98–102]. This study attempted to assess the potential combined impact of the aforementioned factors on the dispersion, aerosolization, and

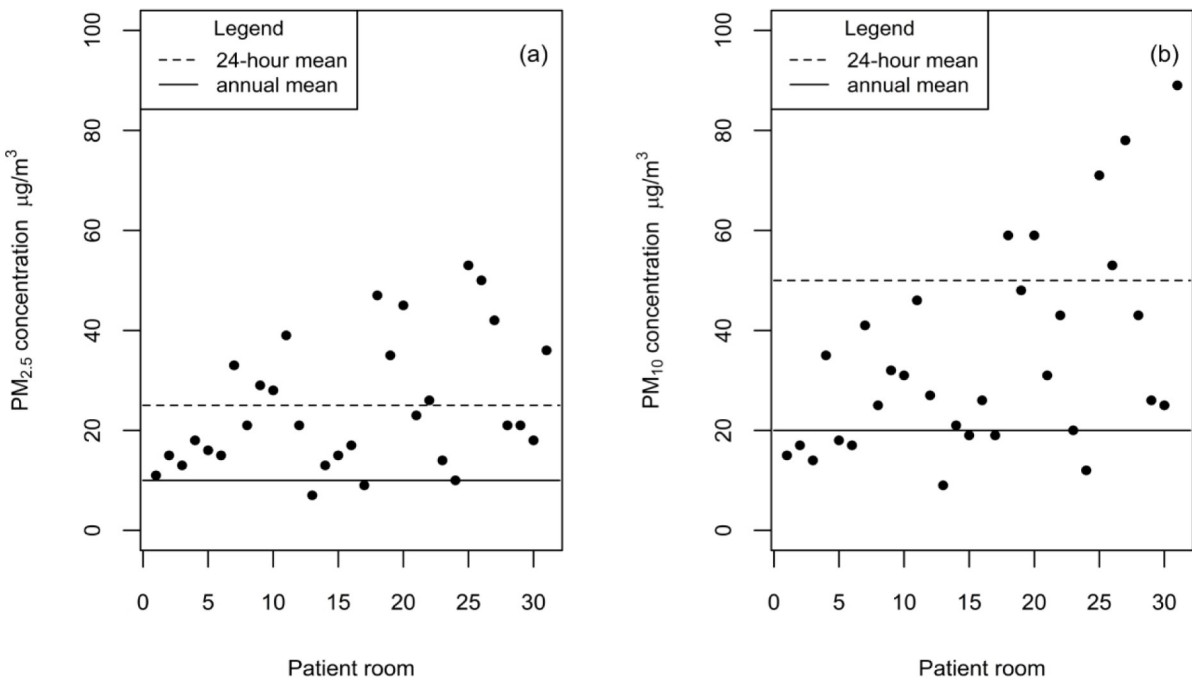

**Fig 3.** Patient room measurements as compared to the 24-hour and annual mean WHO guidelines: (a) $PM_{2.5}$ and (b) $PM_{10}$.

shedding of the influenza virus. The steady-state Gaussian puff model was used to simulate the emission, dispersion and removal of emitted virus and was calibrated using the collected data at 0.5 and 1 m. The calibrated model was used to predict the initial viral load at the source for each patient. Those were found to range between 405 and 20,029 RNA copies/$m^3$ across the 5 patient rooms where the model was run. Note that previous studies reported that the shedding rates vary significantly across patients and are a function of specific patient characteristics such as sex, influenza vaccination date, smoking habits, antiviral medication, BMI (Body Mass Index), and body temperature [103]. Nonetheless, we did not have access to this patient

**Table 2. Predicted versus observed viral load along with estimated shedding rates and viral concentrations 0.1 m from patient.**

| Room Number | Distance (m) | Normalized Measured Concentration (RNA[b] copies/$m^3$) | Predicted Normalized Concentration (RNA copies/$m^3$) | Percent Difference (%)a | Estimated Shedding Rate (RNA copies/cough) | Estimated Concentration at Source (RNA copies/$m^3$) |
|---|---|---|---|---|---|---|
| **617** | 0.5 | 339.4 | 346.3 | 2 | 38 | 12,447 |
| | 1 | 100.3 | 59.5 | -40 | | |
| **526** | 0.5 | 56.2 | 58.7 | 4.5 | 39 | 1,902.8 |
| | 1 | 24.9 | 10.1 | -59 | | |
| **915** | 0.5 | 264.7 | 294.2 | 11.1 | 5.84 | 9,533.6 |
| | 1 | 221.9 | 50.6 | -76 | | |
| **1012** | 0.5 | 11.6 | 12.5 | 7.4 | 9.8 | 405.08 |
| | 1 | 7.4 | 2.2 | -71 | | |
| **1030** | 1 | 358.4 | 349.0 | 2.6 | 24 | 20,029 |
| | 1.5 | 43.8 | 83.0 | 88 | | |

[a] Percentage difference calculated between normalized measured concentration and predicted normalized concentration

[b] Ribonucleic Acid

information in our study and therefore we did not account for them in our analysis. Overall, our model predicted viral loads at 0.5 m better than those at 1 m. Concentrations at 1 m tended to be over-estimated, while those at 0.5 m exhibited an accuracy margin of 10% (Table 2).

The calibrated model was used to compare between the relative magnitudes of the different viral removal pathways (Fig 4). Three major mechanisms that affect the viral loads in air samples were considered, namely dispersion, settling, and ventilation (AER). Dispersion had the highest impact in reducing the concentration moving away from the patient. The model predicted that dispersion alone was able to reduce the virus concentrations by 11.30% in the initial 0.5 m and then by an additional 73.45% by 1 m, yielding a total reduction of 84.76%. Interestingly, the impact of dispersion was more pronounced in the second half-meter as compared to the first half. This can be explained by the typical dispersion pattern of a cough (Fig 5) [35, 79, 81]. Settling, which is mainly affected by T and RH, came second and was responsible for a reduction of $\sim$13.5% over the first 1 m distance. Although the effect of the AER appears to be marginal on the instantaneous reduction of influenza virus levels (Fig 4), this effect can become dominant over an extended period of time if the AER is increased sufficiently [74]. Excluding dispersion and limiting the analysis to the AER and settling, the importance of the AER becomes more notable with a removal efficiency of 9%, and can increase to 13% if the AER is doubled.

A one-at-a-time sensitivity analysis was conducted to assess the impact of changing ambient parameters (RH, T, AER) on virus removal efficiency. The model showed no sensitivity to variations in T over the tested range of [20.6–25.4°C]. Conversely, increasing RH (from 38% to 62%) was found to increase settling efficiency. Note that the impact of RH on the removal rate is mainly due to the transformation of droplets, which takes place right after the shedding point. Depending on the RH of the room, the droplets moisture content decreases, affecting its size and as a result, its settling velocity. As such, at higher RH levels the coughed droplets retain their moisture content and are thus heavier and tend to settle better. Conversely, in less damp environments the droplets tend to lose their moisture content to the ambient air, leading to smaller and lighter particles that can remain suspended in air for longer time. Nevertheless, we found that the model's sensitivity to changes in RH was low (+1.5% change in removal rate for a 10% increase in RH). This could probably explain the lack of a statistically significant correlation between RH and the viral concentrations measured in the rooms. Finally, the model was found to have a low sensitivity to AER when the latter was changed from 6 to 12 ACH. The overall removal efficiency improved by 0.12% for every 1 AER increase. Note that while increasing the AER is expected to improve the removal rate because of improving air recirculation, it can also accelerate dispersion in a patient room due to increased lateral air velocities.

It is important to highlight that the adopted model has several simplifying assumptions and limitations. One of the main limitations of the model was the need to calibrate it using only two samples taken consecutively in each patient room, which limits the confidence in the decay curves that were developed to track virus concentration development over distance. However, as mentioned before, samples were collected consecutively at each distance to avoid interference by running two Coriolis μ Biological Air Sampler concurrently in one small space. Furthermore, the model assumed that the AER and the lateral diffusivity terms were uniform across all rooms. Similarly, the model assumed that the emitted droplet distribution did not vary by patient. Future work should aim towards taking multiple measurements per patient room. Moreover, in this study, coughing was considered to be the main source of the influenza virus and the results were normalized to one cough. However, patients can also shed influenza virus during tidal breathing [103–105]. In addition, future studies may attempt collecting patient specific information pending IRB approval (e.g., age, BMI, medical treatment etc.) in order to better understand patient factors that affect shedding rates and coughing speeds.

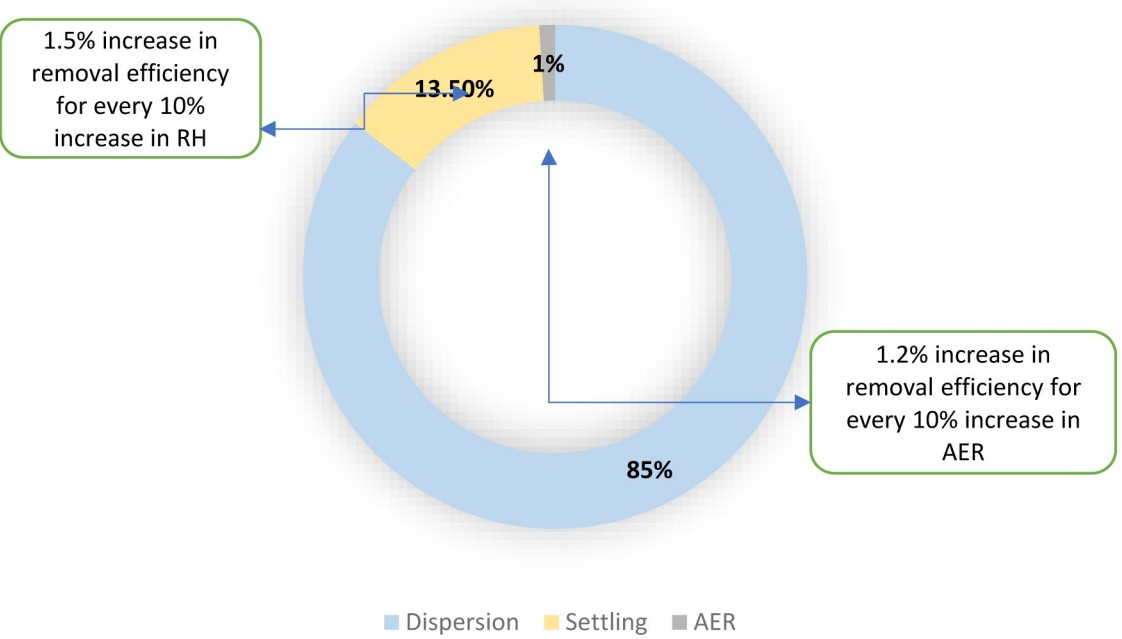

**Fig 4. Virus removal efficiency and mechanisms.**

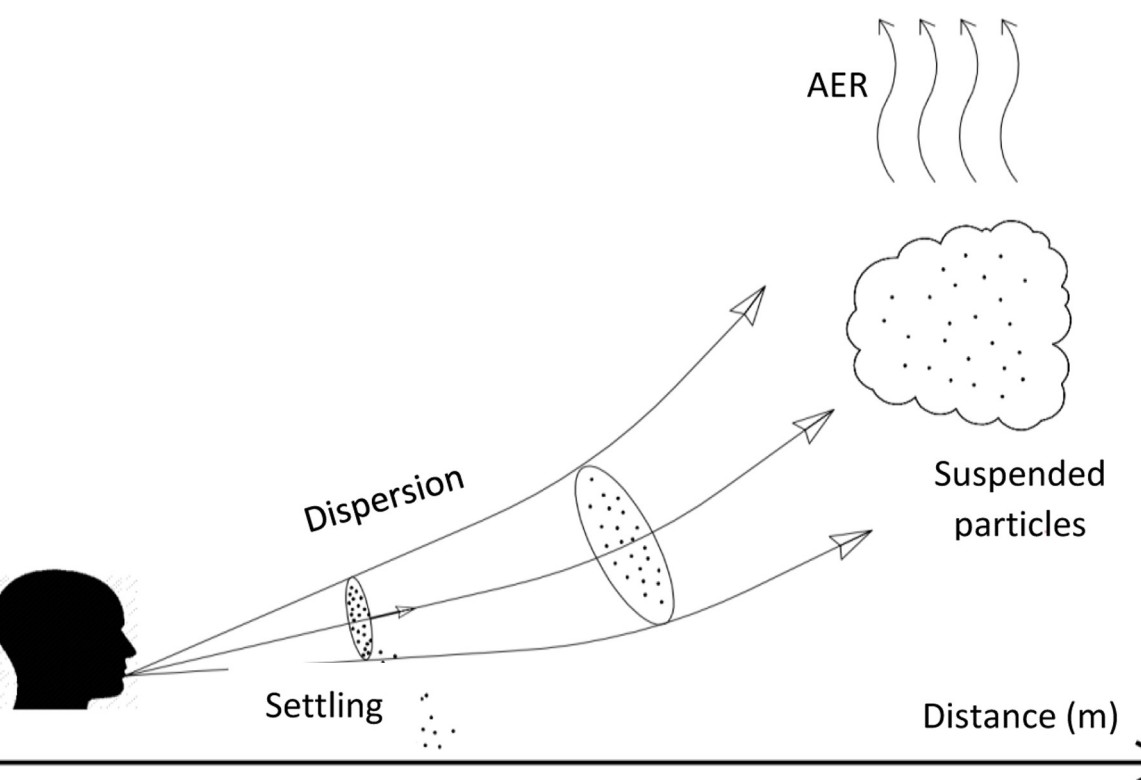

**Fig 5. Horizontal physical representation of the evolution of a cough cloud.**

Finally, the model will benefit from incorporating biological decay as a function of T and RH, which can be incorporated by quantifying the infectious virus load instead of using the viral gene copy number.

This study highlighted the importance of complementing infection control measures with a well-trained facility team that can implement an integrated approach towards IAQ management. Undoubtedly, similar to the modeling efforts, the COVID-19 pandemic has equally pushed the envelope towards adopting IAQ control measures to protect healthcare stakeholders, irrespective of the routes of transmission caused by the high virus infectivity. In this context, besides distancing, masks and cleaning/disinfection [31, 106, 107], ventilation with proper filtration remains the most effective approach for IAQ protection in hospitals albeit the large physical footprint of filtration systems [29, 31, 62, 95, 106, 108–110]. In fact, heating, ventilation, and air conditioning (HVAC) systems have historically been widely used in hospitals to ensure comfort, relieve some temperature-related symptoms, and remove bioaerosols [111]. However, the high infectivity of viruses raised concerns that existing HVAC systems may increase the risk of airborne disease particularly if recirculation systems are not equipped with proper filtration control [31, 96, 112] such as high efficiency particulate air (HEPA) filters [106, 113] that can capture submicron particles [31, 107, 114, 115].

## Conclusion

In this study, airborne viruses, fine particulate matter, air exchange rate, temperature, and relative humidity were monitored in hospital rooms of patients with confirmed infections to identify correlations and assess virus emissions and transport. The results raise concerns about IAQ in healthcare facilities and the potential exposure of staff and visitors to aerosolized viruses as well as elevated indoor PM levels caused by outdoor sources and/or re-suspension of settled particles by indoor activities. The influenza A virus was detected in 14 out of 33 patient rooms with 63% of the positive samples collected at 1 m away from the patient and detectable up to 1.5 m from the patient with a tendency to decrease in magnitude at greater distances away from the patient. $PM_{2.5}$ and $PM_{10}$ daily exposure guidelines for indoor air quality were exceeded in most rooms. While changes in the relative humidity and the room's temperature were found to affect the viral removal efficiency, dispersion caused by the room air exchange rate was a dominant pathway for viral removal.

## Supporting information

**S1 File.**
(DOCX)

## Acknowledgments

Special thanks are extended Dar Al-Handasah (Shair & Partners) endowment for its support to the graduate programs in Engineering at the American University of Beirut. The study was approved by the Institutional Review Board (IRB) at the American University of Beirut Medical Center. All patients provided written informed consents prior to indoor air sample collection. As stipulated in the IRB approval, the research team did not recruit participants. The team was only informed of the date a patient was diagnosed with respiratory syncytial virus (RSV) or influenza A/B. Otherwise, the healthcare team kept the identity of patients confidential. Activities of the research team were limited to in-room monitoring of target indicators using well-defined and declared equipment with no obstruction of routine medical care procedures and hospital protocols. All data generated or analyzed during this study are included in

this article. The authors declare that they have no known conflict of interest that could influence the work reported in this paper.

## Author Contributions

**Conceptualization:** Hassan Zaraket, Mutasem El-Fadel.

**Data curation:** Francis Hanna, Habib Alkalamouni.

**Formal analysis:** Francis Hanna, Ibrahim Alameddine, Hassan Zaraket, Habib Alkalamouni, Mutasem El-Fadel.

**Funding acquisition:** Mutasem El-Fadel.

**Investigation:** Francis Hanna, Habib Alkalamouni.

**Methodology:** Ibrahim Alameddine, Hassan Zaraket, Mutasem El-Fadel.

**Project administration:** Mutasem El-Fadel.

**Resources:** Mutasem El-Fadel.

**Software:** Francis Hanna, Ibrahim Alameddine.

**Supervision:** Mutasem El-Fadel.

**Validation:** Francis Hanna, Ibrahim Alameddine, Hassan Zaraket, Habib Alkalamouni, Mutasem El-Fadel.

**Visualization:** Francis Hanna, Ibrahim Alameddine, Hassan Zaraket, Habib Alkalamouni, Mutasem El-Fadel.

**Writing – original draft:** Francis Hanna.

**Writing – review & editing:** Ibrahim Alameddine, Hassan Zaraket, Habib Alkalamouni, Mutasem El-Fadel.

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
