## [Decision Letter · Decision Letter 0]

31 May 2023

PONE-D-23-10986

Airborne virus shedding by patients in health care units: Removal mechanisms affecting virus transmission

PLOS ONE

Dear Dr. El-Fadel,

Thank you for submitting your manuscript to PLOS ONE. After careful consideration, we feel that it has merit but does not fully meet PLOS ONE’s publication criteria as it currently stands. Therefore, we invite you to submit a revised version of the manuscript that addresses the points raised during the review process.

We look forward to receiving your revised manuscript.

Kind regards,

Sreekumar Othumpangat, PhD

Academic Editor

PLOS ONE

Journal Requirements:

Detection of influenza virus in air samples of patient rooms - https://doi.org/10.1016/j.jhin.2020.10.020

In your revision ensure you cite all your sources (including your own works), and quote or rephrase any duplicated text outside the methods section. Further consideration is dependent on these concerns being addressed.

Additional Editor Comments:

Both reviewers are of the impression that the title of the manuscript be changed to avoid any ambiguity and remove the RSV part. One of the reviewers has given a very detailed suggestions to improve the manuscript quality and I suggest the authors to give good attention to the critical comments when modifying the manuscript..If possible give patients demographic information for  better understanding of the data presented.

Reviewers' comments:

Reviewer's Responses to Questions

**Comments to the Author**

1. Is the manuscript technically sound, and do the data support the conclusions?

Reviewer #1: Partly

Reviewer #2: Yes

2. Has the statistical analysis been performed appropriately and rigorously? 

Reviewer #1: I Don't Know

Reviewer #2: Yes

3. Have the authors made all data underlying the findings in their manuscript fully available?

Reviewer #1: Yes

Reviewer #2: Yes

4. Is the manuscript presented in an intelligible fashion and written in standard English?

Reviewer #1: Yes

Reviewer #2: Yes

5. Review Comments to the Author

Reviewer #1: The paper reports data on airborne influenza concentrations and other parameters collected in a hospital, and presents a model of the effects of different parameters on airborne influenza concentrations. Although previous studies have examined airborne influenza in healthcare facilities and other locations, the data in the paper is useful. However, I have concerns about the reliability of the model.

The title says “airborne virus” and the first sentence of the abstract says, “In this study, we characterize the distribution of airborne viruses (respiratory syncytial virus (RSV) and influenza A/B) in hospital rooms of patients with confirmed infections.” However, none of the patients in the study has RSV, and the only viruses that were studied were influenza A and B. The title should be changed to something like “airborne influenza virus” and the abstract rewritten to just say “airborne influenza A and B viruses”. This also needs to be corrected in the last paragraph of the Introduction and elsewhere throughout the paper.

In the Introduction, the authors say, “the avian influenza virus”. This should be avian influenza viruses, since there are numerous strains. Later in the same sentence the authors say, “COVID-19 pandemic [32-34] caused by the severe acute respiratory syndrome coronavirus (SARS-CoV1 or SARS-CoV2).” COVID-19 is caused by SARS-CoV-2, not SARS-CoV-1. Note also that the virus is written as SARS-CoV-2, not SARS-CoV2.

Introduction, second paragraph: The authors are correct that, historically, droplets have been defined as particles > 5 µm in diameter that fall rapidly to the ground, while droplet nuclei (aerosols) were defined as particles < 5 µm that can remain airborne for an extended period. In the past, these definitions were used to distinguish between airborne transmission by aerosols and droplet transmission by droplets. However, these definitions are not based on aerosol science, and in reality particles much larger than 5 µm can remain airborne for a significant time and be inhaled. Unfortunately, during the early part of the COVID-19 pandemic, these definitions were used to justify the erroneous assumption that COVID-19 was not spread by aerosols. Humans continuously emit aerosols of respiratory fluids as they breathe, talk, cough, sneeze, and sing. These respiratory aerosols can have a very broad size range, from tens of nanometers in diameter to visible droplets of a millimeter or more (Morawska et al. 2009; Gralton et al. 2011; Bourouiba et al. 2014; Fennelly 2020). Airborne particles larger than 100 µm are ballistic; that is, they are affected primarily by gravity and fall quickly to the ground. Respiratory aerosol particles in this size range tend to deposit within a few meters of the source (Prather et al. 2020). As the aerosol particle diameter decreases from 100 µm, a gradual transition occurs where the settling velocity rapidly decreases and the particles remain airborne for longer times. For example, a 100 µm aerosol particle takes 4 seconds to fall 1 meter in still air, while a 10 µm aerosol particle takes 5.4 minutes, a 5 µm aerosol particle takes 21 minutes, and a 1 µm aerosol particle takes 8 hours to settle the same distance (Hinds 1999). Because of this, there is a growing consensus in the aerosol science community that the term aerosols and aerosol transmission should be used for particles less than 100 µm rather than 5 µm, and that droplet transmission be used only for particles larger than 100 µm that truly do settle rapidly. See these references: (Fennelly 2020; Prather et al. 2020; Marr and Tang 2021; Randall et al. 2021; Tang et al. 2021; Jimenez et al. 2022)

Introduction, second paragraph: The authors say, “Variations in temperature (T) and relative humidity (RH) play an important role in the activation or deactivation of viruses [39-41]”. Viruses are expelled from the body and become deactivated over time; they are not activated by the environment.

Introduction, third paragraph: The authors say, “Despite earlier efforts and until recently, many of these studies fell short of defining clear statistical relationships between the concentrations of the influenza virus (or other microorganisms) in patient rooms and the physical determinants (such as T, RH, AER) and the location of the patient in the room [48].” I would expect that the most fundamental factor determining the concentration of airborne influenza virus would be the rate of airborne virus shedding by the patient.

Introduction, third paragraph: The authors say, “In this study, we monitor the viral load (RSV and influenza A/B), particulate matter (PM2.5 & PM10,), and physical parameters (AER, T, RH) within patient rooms for the purpose of assessing airborne virus shedding by patients in health care units using an integrated approach of coupling field measurements with statistical and numerical analyses.” Why would you expect there to be a relationship between PM and airborne virus? These would seem to be independent to me, unless PM is an indirect indicator of room ventilation and filtration or something similar.

Materials and Methods, second paragraph: The Coriolis sampler only collects aerosol particles from about 0.5 to 20 µm. The authors should note this.

Figure 1: More information is needed here. Where was the ventilation system inlet into the room and outlet from the room relative to the patient and aerosol samplers? The amount of virus collected could be greatly affected if the samples were collected directly downwind or upwind from the patient.

Materials and Methods, equations 2-5: This seem like a very complex model with a large number of assumptions that are being applied to a very limited data set. I question how reliable the results are here. Note, for example, that the authors have assumed that the surface tension of the droplets is 0.072 N/m because that is the surface tension of pure water. In fact, however, this is not true: respiratory fluids contain a large amount of surfactant and the surface tension of these fluids is much lower than water.

Materials and Methods, page 9: The recitation of variables here in paragraph form is very difficult to read. The variable should be given as a list with one line for each variable.

Materials and Methods, page 10: The authors say, “The AER was estimated to be 7.2 AER and was validated by the hospital’s physical plant engineers.” Was the AER of the study rooms actually measured during the study? The nominal setting for the AER can vary from the actual AER, and the AER can vary from room to room. How confident are the authors that this value was correct for all the rooms?

Materials and Methods, page 11: The authors say, “Normalization was conducted by dividing the total concentration measures in the sample by the number of coughs recorded over the sampling period.” Patients with influenza shed virus during tidal breathing as well as coughing, and some researchers believe that tidal breathing is actually a greater source of virus than coughing for most patients. See, for example, (Milton et al. 2013; Yan et al. 2018; Bueno de Mesquita et al. 2021)

Results and Discussion, page 18: The authors say, “Moreover, the role of AER on removal was found to be marginal with a reduction rate of 1%”. I see several problems with this statement. The authors did not vary the AER during their study, so their data does not support this conclusion. The authors’ model assumes a value for the combined effects of settling and ventilation; it is not based on their experimental results. Numerous studies have shown that increased ventilation is an effective way to reduce exposure to airborne particles. In fact, the Yang and Marr paper that the authors cite as the basis for the assumed decay rate in their model (Yang and Marr 2011) says, “We have demonstrated the relative importance of the three removal mechanisms. Settling can remove over 80% of droplets emitted from a cough within 10 min; however, it is effective only for larger droplets and allows the smaller ones (<5 µm) to remain suspended. In contrast, ventilation is able to remove all droplets regardless of size simply by air exchange. Therefore, higher AERs will facilitate the elimination of virus-containing droplets from indoor environments, especially to compensate for the inefficacy of settling in removing the small ones. This observation also justifies the requirement to maintain a high AER in public places (e.g., 12 ACH in hospital waiting areas).”

References

Bourouiba, L, E Dehandschoewercker and John WM Bush (2014). Violent expiratory events: on coughing and sneezing. J Fluid Mech 745: 537-563. https://doi.org/10.1017/jfm.2014.88

Bueno de Mesquita, PJ, J Nguyen-Van-Tam, B Killingley, J Enstone, R Lambkin-Williams, AS Gilbert, A Mann, J Forni, J Yan, J Pantelic, ML Grantham and DK Milton (2021). Influenza A (H3) illness and viral aerosol shedding from symptomatic naturally infected and experimentally infected cases. Influenza Other Respir Viruses 15(1): 154-163. https://doi.org/10.1111/irv.12790

Fennelly, KP (2020). Particle sizes of infectious aerosols: implications for infection control. Lancet Respir Med 8(9): 914-924. https://doi.org/10.1016/S2213-2600(20)30323-4

Gralton, J, E Tovey, ML McLaws and WD Rawlinson (2011). The role of particle size in aerosolised pathogen transmission: a review. J Infect 62(1): 1-13. https://doi.org/10.1016/j.jinf.2010.11.010

Hinds, WC (1999). Aerosol Technology. Properties, Behavior, and Measurement of Airborne Particles. New York, John Wiley & Sons.

Jimenez, JL, LC Marr, K Randall, ET Ewing, Z Tufekci, T Greenhalgh, R Tellier, JW Tang, Y Li, L Morawska, J Mesiano-Crookston, D Fisman, O Hegarty, SJ Dancer, PM Bluyssen, G Buonanno, M Loomans, WP Bahnfleth, M Yao, C Sekhar, P Wargocki, AK Melikov and KA Prather (2022). What were the historical reasons for the resistance to recognizing airborne transmission during the COVID-19 pandemic? Indoor Air 32(8): e13070. https://doi.org/10.1111/ina.13070

Marr, LC and JW Tang (2021). A Paradigm Shift to Align Transmission Routes With Mechanisms. Clin Infect Dis 73(10): 1747-1749. https://doi.org/10.1093/cid/ciab722

Milton, DK, MP Fabian, BJ Cowling, ML Grantham and JJ McDevitt (2013). Influenza virus aerosols in human exhaled breath: particle size, culturability, and effect of surgical masks. PLoS Pathog 9(3): e1003205. https://doi.org/10.1371/journal.ppat.1003205

Morawska, L, GR Johnson, ZD Ristovski, M Hargreaves, K Mengersen, S Corbett, CYH Chao, Y Li and D Katoshevski (2009). Size distribution and sites of origin of droplets expelled from the human respiratory tract during expiratory activities. J Aerosol Sci 40(3): 256-269. https://doi.org/10.1016/j.jaerosci.2008.11.002

Prather, KA, LC Marr, RT Schooley, MA McDiarmid, ME Wilson and DK Milton (2020). Airborne transmission of SARS-CoV-2. Science 370(6514): 303-304. https://doi.org/10.1126/science.abf0521

Randall, K, ET Ewing, LC Marr, JL Jimenez and L Bourouiba (2021). How did we get here: what are droplets and aerosols and how far do they go? A historical perspective on the transmission of respiratory infectious diseases. Interface Focus 11(6): 20210049. https://doi.org/10.1098/rsfs.2021.0049

Tang, JW, WP Bahnfleth, PM Bluyssen, G Buonanno, JL Jimenez, J Kurnitski, Y Li, S Miller, C Sekhar, L Morawska, LC Marr, AK Melikov, WW Nazaroff, PV Nielsen, R Tellier, P Wargocki and SJ Dancer (2021). Dismantling myths on the airborne transmission of severe acute respiratory syndrome coronavirus-2 (SARS-CoV-2). J Hosp Infect 110: 89-96. https://doi.org/10.1016/j.jhin.2020.12.022

Yan, J, M Grantham, J Pantelic, PJ Bueno de Mesquita, B Albert, F Liu, S Ehrman, DK Milton and E Consortium (2018). Infectious virus in exhaled breath of symptomatic seasonal influenza cases from a college community. Proc Natl Acad Sci USA 115(5): 1081-1086. https://doi.org/10.1073/pnas.1716561115

Yang, W and LC Marr (2011). Dynamics of airborne influenza A viruses indoors and dependence on humidity. PLoS ONE 6(6): e21481. https://doi.org/10.1371/journal.pone.0021481

Reviewer #2: This is a very well written paper with a sound scientific approach and appropriate conclusions. The technical and mathematical descriptions may be difficult for non-specialized readers but this is forgivable. The conclusions are of wide impact and should be transmitted to different disciplines including infection control programs, hospital engineers, and infectious diseases experts. Following are some comments and suggestions:

• The patient population is not described. Were all the subjects enrolled adult patients? Any pediatric patients? Were there any elderly patients who may have a weakened cough? Did any patients receive nebulized medications or saline as part of their treatment? If this information was not available, it should be stated as a limitation

• Since all the patients enrolled had influenza virus and none had RSV, I wonder why the authors refer to “RSV” in their abstract and use the title as “airborne virus” instead of “influenza”. I think the latter would be more appropriate.

• A brief description of the Coriolis µ Biological Air Sampler should be provided. How it works, volume collected, filters, etc.

• Lines 79-83: It is not clear how written informed consent was obtained (where by definition the patient has to sign their name), yet the patients’ identities were kept confidential. This is a bit confusing.

• Line 172: the unit for AER is 1/hour or L/hour?

• Lines 187-188: “The AER was estimated to be 7.2 AER…”. Is there a unit missing or a period after “7.2”?

• Figure 2: The labeling of the pie-charts should be corrected

6. PLOS authors have the option to publish the peer review history of their article (what does this mean?). If published, this will include your full peer review and any attached files.

Reviewer #1: No

Reviewer #2: No

---

## [Author Response · Author response to Decision Letter 0]

16 Jul 2023

Response to Reviewers’ Comments

Manuscript Number: PONE-D-23-10986

Airborne virus shedding by patients in health care units:

Removal mechanisms affecting virus transmission

Reviewer #1

The paper reports data on airborne influenza concentrations and other parameters collected in a hospital, and presents a model of the effects of different parameters on airborne influenza concentrations. Although previous studies have examined airborne influenza in healthcare facilities and other locations, the data in the paper is useful. However, I have concerns about the reliability of the model.

We appreciate the valuable comments of the reviewer and made every effort to address them as outlined below. These comments have indeed helped us in improving the paper.

Comment 1) The title says “airborne virus” and the first sentence of the abstract says, “In this study, we characterize the distribution of airborne viruses (respiratory syncytial virus (RSV) and influenza A/B) in hospital rooms of patients with confirmed infections.” However, none of the patients in the study has RSV, and the only viruses that were studied were influenza A and B. The title should be changed to something like “airborne influenza virus” and the abstract rewritten to just say “airborne influenza A and B viruses”. This also needs to be corrected in the last paragraph of the Introduction and elsewhere throughout the paper.

Response 1) We thank the reviewer for the suggestion. We agree with the comment. All reference to RSV were removed. Changes were made to the Title, Abstract, Introduction, and the rest of the revised manuscript (lines 1, 14, 38, 73, 77, 84, 246, 248). 

Comment 2) In the Introduction, the authors say, “the avian influenza virus”. This should be avian influenza viruses, since there are numerous strains. Later in the same sentence the authors say, “COVID-19 pandemic [32-34] caused by the severe acute respiratory syndrome coronavirus (SARS-CoV1 or SARS-CoV2).” COVID-19 is caused by SARS-CoV-2, not SARS-CoV-1. Note also that the virus is written as SARS-CoV-2, not SARS-CoV2.

Response 2) We thank the reviewer for the insightful comment. The proposed changes were made in the revised manuscript (line 43).

Comment 3) Introduction, second paragraph: The authors are correct that, historically, droplets have been defined as particles > 5 µm in diameter that fall rapidly to the ground, while droplet nuclei (aerosols) were defined as particles < 5 µm that can remain airborne for an extended period. In the past, these definitions were used to distinguish between airborne transmission by aerosols and droplet transmission by droplets. However, these definitions are not based on aerosol science, and in reality particles much larger than 5 µm can remain airborne for a significant time and be inhaled. Unfortunately, during the early part of the COVID-19 pandemic, these definitions were used to justify the erroneous assumption that COVID-19 was not spread by aerosols. Humans continuously emit aerosols of respiratory fluids as they breathe, talk, cough, sneeze, and sing. These respiratory aerosols can have a very broad size range, from tens of nanometers in diameter to visible droplets of a millimeter or more (Morawska et al. 2009; Gralton et al. 2011; Bourouiba et al. 2014; Fennelly 2020). Airborne particles larger than 100 µm are ballistic; that is, they are affected primarily by gravity and fall quickly to the ground. Respiratory aerosol particles in this size range tend to deposit within a few meters of the source (Prather et al. 2020). As the aerosol particle diameter decreases from 100 µm, a gradual transition occurs where the settling velocity rapidly decreases and the particles remain airborne for longer times. For example, a 100 µm aerosol particle takes 4 seconds to fall 1 meter in still air, while a 10 µm aerosol particle takes 5.4 minutes, a 5 µm aerosol particle takes 21 minutes, and a 1 µm aerosol particle takes 8 hours to settle the same distance (Hinds 1999). Because of this, there is a growing consensus in the aerosol science community that the term aerosols and aerosol transmission should be used for particles less than 100 µm rather than 5 µm, and that droplet transmission be used only for particles larger than 100 µm that truly do settle rapidly. See these references: (Fennelly 2020; Prather et al. 2020; Marr and Tang 2021; Randall et al. 2021; Tang et al. 2021; Jimenez et al. 2022)

Response 3) We thank the reviewer for providing such an informative summary on the distinction between biological droplets and aerosols. We also thank him/her for listing some of the seminal work in the field of aerosol and droplet transmission. We have attempted to summarize the provided information in the revised introduction (lines 44-52).

Comment 4) Introduction, second paragraph: The authors say, “Variations in temperature (T) and relative humidity (RH) play an important role in the activation or deactivation of viruses [39-41]”. Viruses are expelled from the body and become deactivated over time; they are not activated by the environment.

Response 4) The sentence has been revised (line 53).

Comment 5) Introduction, third paragraph: The authors say, “Despite earlier efforts and until recently, many of these studies fell short of defining clear statistical relationships between the concentrations of the influenza virus (or other microorganisms) in patient rooms and the physical determinants (such as T, RH, AER) and the location of the patient in the room [48].” I would expect that the most fundamental factor determining the concentration of airborne influenza virus would be the rate of airborne virus shedding by the patient.

Response 5) We agree with the reviewer that the shedding rate is indeed the major factor that affects the concentration of airborne influenza virus. In the sentence in question, we intended to emphasize the impact of factors that are associated with the physical environment within a patient room. We revised the sentence to clarify the point (lines 63-65).

Comment 6) Introduction, third paragraph: The authors say, “In this study, we monitor the viral load (RSV and influenza A/B), particulate matter (PM2.5 & PM10,), and physical parameters (AER, T, RH) within patient rooms for the purpose of assessing airborne virus shedding by patients in health care units using an integrated approach of coupling field measurements with statistical and numerical analyses.” Why would you expect there to be a relationship between PM and airborne virus? These would seem to be independent to me, unless PM is an indirect indicator of room ventilation and filtration or something similar.

Response 6) We agree with the reviewer that the relationship might only be a correlation (with a common latent variable) rather than causal. The link between air quality parameters and IAV remains an open research area. For example, Su et al., (2019) reported a positive correlation between PM and the influenza virus; meanwhile, Nor et al., 2021 reported no direct link between PM and SARS-CoV-2. In this study, no statistically significant correlation between PM and the influenza virus was found. The manuscript was revised accordingly (lines 287-290).

References

Nor, N.S.M., Yip, C.W., Ibrahim, N., Jaafar, M.H., Rashid, Z.Z., Mustafa, N., Hamid, H.H.A., Chandru, K., Latif, M.T., Saw, P.E., Lin, C.Y., Alhasa, K.M., Hashim, J.H. and Nadzir, M.S.M., 2021. Particulate matter (PM(2.5)) as a potential SARS-CoV-2 carrier. Sci Rep, 11(1): 2508.

Su, W., Wu, X., Geng, X., Zhao, X., Liu, Q. and Liu, T., 2019. The short-term effects of air pollutants on influenza-like illness in Jinan, China. BMC Public Health, 19(1): 1319.

Comment 7) Materials and Methods, second paragraph: The Coriolis sampler only collects aerosol particles from about 0.5 to 20 µm. The authors should note this.

Response 7) We thank the reviewer for the comment. This has been noted in the revised manuscript (line 112).

Comment 8) Figure 1: More information is needed here. Where was the ventilation system inlet into the room and outlet from the room relative to the patient and aerosol samplers? The amount of virus collected could be greatly affected if the samples were collected directly downwind or upwind from the patient.

Response 8) The inlet-outlet system are located at the ceiling of the entrance door and in the toilet within the patient’s room. The sampling locations were selected to minimize the impact of the ventilation system inflow/outflow. This has been noted in the revised manuscript with an updated Figure 1 showing the location of the inflow outflow system (lines 105-106; and Figure 1).

Comment 9) Materials and Methods, equations 2-5: This seem like a very complex model with a large number of assumptions that are being applied to a very limited data set. I question how reliable the results are here. Note, for example, that the authors have assumed that the surface tension of the droplets is 0.072 N/m because that is the surface tension of pure water. In fact, however, this is not true: respiratory fluids contain a large amount of surfactant and the surface tension of these fluids is much lower than water.

Response 9) We agree that models can be relatively complex and might be affected by over-parameterization. The surface tension for coughed droplets was initially set at 0.072 mN/m, the same as pure water, to reduce the number of calibration parameters. To assess the sensitivity of the results to changes in the surface tension, we varied this value based on the recent work of Vadlamudi et al., (2022) who reported a surface tension of 0.0659 mN/m for coughed droplets. The results showed that within this range, changes to the surface tension value exhibited a negligible effect on the equilibrium diameter, and thus, a negligible effect on the settling efficiency. The remaining parameters in Equation (3) were assigned values reported in the literature (i.e., molar mass, gas constant, chemical composition of saliva). We clarified these points in the revised manuscript (lines 210-216).

References:

Vadlamudi, G., Thirumalaikumaran, S.K., Chakravortty, D., Saha, A. and Basu, S., 2022. Penetration and aerosolization of cough droplet spray through face masks: A unique pathway of transmission of infection. Physics of Fluids, 34(5). 

Comment 10) Materials and Methods, page 9: The recitation of variables here in paragraph form is very difficult to read. The variable should be given as a list with one line for each variable.

Response 10) The format of the variables was changed in the revised manuscript as suggested (lines 137-140, 162-189).

Comment 11) Materials and Methods, page 10: The authors say, “The AER was estimated to be 7.2 AER and was validated by the hospital’s physical plant engineers.” Was the AER of the study rooms actually measured during the study? The nominal setting for the AER can vary from the actual AER, and the AER can vary from room to room. How confident are the authors that this value was correct for all the rooms?

Response 11) The AERs in this study could not be measured in each room due to imposed constraints in the presence of patients. AERs were calculated based on the air intake and the volume of each room (Equation (6)). In earlier work, the AER was measured in several similar rooms within the same hospital using a TSI ACCUBALANCE Air Capture Hood Model 8380 (Chamseddine, 2018). The measurements were then verified with the hospital’s physical plant engineers. While the measurements occurred in a different time period, they provided confidence that the estimated AERs were close to those measured. The manuscript was revised accordingly (lines 203-209)

Reference

Chamseddine, A., 2018. Determinants of Indoor Air Quality in Hospitals: Impact of ventilation systems with Indoor-Outdoor correlations and health implications. PhD Dissertation, Department of Civil and Environmental Engineering, American University of Beirut.

Comment 12) Materials and Methods, page 11: The authors say, “Normalization was conducted by dividing the total concentration measures in the sample by the number of coughs recorded over the sampling period.” Patients with influenza shed virus during tidal breathing as well as coughing, and some researchers believe that tidal breathing is actually a greater source of virus than coughing for most patients. See, for example, (Milton et al. 2013; Yan et al. 2018; Bueno de Mesquita et al. 2021)

Response 12) This has been noted in the revised manuscript (lines 363-366). The aim of normalization was to account for discrepancies observed in the number of recorded coughs across patients.

Comment 13) Results and Discussion, page 18: The authors say, “Moreover, the role of AER on removal was found to be marginal with a reduction rate of 1%”. I see several problems with this statement. The authors did not vary the AER during their study, so their data does not support this conclusion. The authors’ model assumes a value for the combined effects of settling and ventilation; it is not based on their experimental results. Numerous studies have shown that increased ventilation is an effective way to reduce exposure to airborne particles. In fact, the Yang and Marr paper that the authors cite as the basis for the assumed decay rate in their model (Yang and Marr 2011) says, “We have demonstrated the relative importance of the three removal mechanisms. Settling can remove over 80% of droplets emitted from a cough within 10 min; however, it is effective only for larger droplets and allows the smaller ones (<5 µm) to remain suspended. In contrast, ventilation is able to remove all droplets regardless of size simply by air exchange. Therefore, higher AERs will facilitate the elimination of virus-containing droplets from indoor environments, especially to compensate for the inefficacy of settling in removing the small ones. This observation also justifies the requirement to maintain a high AER in public places (e.g., 12 ACH in hospital waiting areas).”

Response 13) While we agree with the reviewer that under extended operational periods, changes to the AER can control virus-containing droplets, the Yang and Marr (2011) study and the current study are slightly different. In this study, we simulated virus shedding and transmission while highlighting the mechanisms affecting the virus concentration at four points: 1) the shedding source, 2) at 0.5 m from the patient, 3) at 1 m from the patient, and 4) 1.5 m away from the patient. The model represents a point in time when the patient is coughing with a focus on the source and 1 minute thereafter. In contrast, in the Yang and Marr (2011) study, the authors focused on the removal efficiency (15-80%) under variable AER (1 to 10 ACH) over a period of 10 minutes. In the same context, the authors assumed no recirculation and recognized the impact of this limitation in overestimating the AER’s efficiency. While the importance of the AER in controlling virus-containing particles over an extended period of time is evident, dispersion and RH can be more effective in reducing the virus concentrations immediately after the shedding process. In this study, although the effect of the AER appears to be marginal on the instantaneous reduction of influenza virus levels (Figure 4), this effect can become dominant over an extended period of time if the AER is increased sufficiently [74]. Excluding dispersion and limiting the analysis to the AER and settling, the importance of the AER becomes more notable with a removal efficiency of 9%, and can increase to 13% if the AER is doubled (always immediately after the shedding process and not for a 10 min period reported in Yang and Marr (2011)).

The manuscript was revised accordingly for more clarity (lines 326-330).

 

Reviewer #2

This is a very well written paper with a sound scientific approach and appropriate conclusions. The technical and mathematical descriptions may be difficult for non-specialized readers but this is forgivable. The conclusions are of wide impact and should be transmitted to different disciplines including infection control programs, hospital engineers, and infectious diseases experts. Following are some comments and suggestions:

We appreciate the valuable comments of the reviewer and made every effort to address them as outlined below. These comments have indeed helped us in improving the paper.

Comment 1- The patient population is not described. Were all the subjects enrolled adult patients? Any pediatric patients? Were there any elderly patients who may have a weakened cough? Did any patients receive nebulized medications or saline as part of their treatment? If this information was not available, it should be stated as a limitation

Response 1- The data presented in this manuscript targeted adult patients only. Moreover, none of the patients was elderly. We did not have access to the patients’ charts and as such we had no knowledge if they had or were receiving any nebulized medications or saline as part of their treatment. We only collected information pertaining to whether they were receiving anti-viral medication. We have clarified these points in the revised manuscript (lines 246-250, 259, 367)

Comment 2- Since all the patients enrolled had influenza virus and none had RSV, I wonder why the authors refer to “RSV” in their abstract and use the title as “airborne virus” instead of “influenza”. I think the latter would be more appropriate.

Response 2- We agree with the suggestion. We made the needed changes to the Title, Abstract, and Introduction. We also made sure that these changes were integrated throughout the revised manuscript (lines 1, 14, 38, 73, 77, 84, 246, 248).

Comment 3- A brief description of the Coriolis µ Biological Air Sampler should be provided. How it works, volume collected, filters, etc.

Response 3- We thank the reviewer for the suggestion. We added more information on the Coriolis µ Biological Air Sampler in the Materials and Methods section of the revised manuscript (lines 90-93, 105-106, 112).

Comment 4- Lines 79-83: It is not clear how written informed consent was obtained (where by definition the patient has to sign their name); yet the patients’ identities were kept confidential. This is a bit confusing.

Response 4- The Institutional Review Board (IRB) at the American University of Beirut Medical Center (AUBMC) classified this study under the category of Human Subject Research. Accordingly, the IRB required approval of the patient. Prior to sampling, the patients were provided with a consent form describing the research objectives, the sampling procedure, the risks and benefits of participating in the study, and privacy and confidentiality of the data. Accordingly, we sampled a room only if the patient signed the consent form. The signature did not necessarily reflect or divulge the identity of the patient. The identities of the patients remained confidential on all records collected. 

Comment 5- Line 172: the unit for AER is 1/hour or L/hour?

Response 5- The AER unit is 1/hour which means one room volume exchanged per hour

Comment 6- Lines 187-188: “The AER was estimated to be 7.2 AER…”. Is there a unit missing or a period after “7.2”?

Response 6- The AER unit is 1/hour or ACH (Air changes per hour). The unit is corrected. We revised the sentence by removing the second AER (line 209).

Comment 7- Figure 2: The labeling of the pie-charts should be corrected

Response 7- Thank you for the comment. The labeling of the pie-chart was corrected in the revised manuscript (page 12, Figure 2).

---

## [Decision Letter · Decision Letter 1]

2 Aug 2023

Airborne influenza virus shedding by patients in health care units: Removal mechanisms affecting virus transmission

PONE-D-23-10986R1

Dear Dr. El-Fadel,

We’re pleased to inform you that your manuscript has been judged scientifically suitable for publication and will be formally accepted for publication once it meets all outstanding technical requirements.

Kind regards,

Sreekumar Othumpangat, PhD

Academic Editor

PLOS ONE

Additional Editor Comments (optional):

Reviewers' comments:

Reviewer's Responses to Questions

**Comments to the Author**

1. If the authors have adequately addressed your comments raised in a previous round of review and you feel that this manuscript is now acceptable for publication, you may indicate that here to bypass the “Comments to the Author” section, enter your conflict of interest statement in the “Confidential to Editor” section, and submit your "Accept" recommendation.

Reviewer #1: All comments have been addressed

Reviewer #2: All comments have been addressed

2. Is the manuscript technically sound, and do the data support the conclusions?

Reviewer #1: Yes

Reviewer #2: Yes

3. Has the statistical analysis been performed appropriately and rigorously? 

Reviewer #1: Yes

Reviewer #2: Yes

4. Have the authors made all data underlying the findings in their manuscript fully available?

Reviewer #1: Yes

Reviewer #2: Yes

5. Is the manuscript presented in an intelligible fashion and written in standard English?

Reviewer #1: Yes

Reviewer #2: Yes

6. Review Comments to the Author

Reviewer #1: All of my comments have been addressed. I don't have anything further to add, but your editorial manager software won't let me submit my review until this box contains at least 100 characters, which is annoying.

Reviewer #2: All comments have been answered. The paper is ready for publication from my end.

I don't have anything more to add

7. PLOS authors have the option to publish the peer review history of their article (what does this mean?). If published, this will include your full peer review and any attached files.

Reviewer #1: No

Reviewer #2: No

---

## [Editor Report · Acceptance letter]

4 Aug 2023

PONE-D-23-10986R1 

Airborne influenza virus shedding by patients in health care units: Removal mechanisms affecting virus transmission 

Dear Dr. El-Fadel:

I'm pleased to inform you that your manuscript has been deemed suitable for publication in PLOS ONE. Congratulations! Your manuscript is now with our production department. 

Kind regards, 

on behalf of

Dr. Sreekumar Othumpangat 

Academic Editor

PLOS ONE